# Religion Involvement and Substance Use Problems in Schoolchildren in Northern Chile

Alejandra Caqueo-Urízar [1,*], Erika Peroza [2], Carolang Escobar-Soler [3], Jerome Flores [3,4], Alfonso Urzúa [5], Matías Irarrázaval [6,7], Karina Reygadas [3] and Gustavo Zegarra [3]

[1] Instituto de Alta Investigación, Universidad de Tarapacá, Arica 1000000, Chile
[2] Hospital Regional Dr. Hernán Henríquez Aravena, Manuel Montt 115, Universidad de la Frontera, Temuco 4780000, Chile; egpm1902@gmail.com
[3] Escuela de Psicología y Filosofía, Facultad de Ciencias Sociales y Jurídicas, Universidad de Tarapacá, Arica 1000000, Chile; cescobars@uta.cl (C.E.-S.); jflores@uta.cl (J.F.); kari_reygadas@hotmail.com (K.R.); gustavozegarrarios@gmail.com (G.Z.)
[4] Centro de Justicia Educacional CJE, Pontificia Universidad Católica de Chile, Santiago 8380453, Chile
[5] Escuela de Psicología, Universidad Católica del Norte, Antofagasta 1270709, Chile; alurzua@ucn.cl
[6] Departamento de Psiquiatría, Facultad de Medicina, Hospital Clínico de la Universidad de Chile, Santiago 8380453, Chile; mirarrazavald@uchile.cl
[7] Institute for Depression and Personality Research MIDAP, Santiago 8380453, Chile
* Correspondence: acaqueo@academicos.uta.cl

**Abstract:** (1) Background: Religious involvement and spirituality have proven to be sources of well-being for individuals at different moments in life and are also associated with a decrease in depression, anxiety, and substance use. Therefore, these could be protective factors against stressful conditions and contribute to mental health. The aim of the present study was to analyze the relationship between religious involvement and substance use among students in northern Chile. (2) Methods: The design is retrospective ex post facto with only one group, and the sample included 2313 adolescents between 12 and 18 years of age from public and private schools. A subscale of the Child and Adolescent Assessment System was used to assess substance use and Universal Age I-E-12 to measure religious involvement. (3) Results: The findings suggest that the intrinsic orientation of religiousness ($\beta = -0.048$, $p < 0.014$), age ($\beta = 0.374$, $p < 0.000$), gender ($\beta = 0.039$, $p < 0.040$), and ethnic identity ($\beta = 0.051$, $p < 0.008$) have significant correlations with substance use. (4) Conclusions: The intrinsic orientation of religion is a relevant variable associated with consumption due to its non-instrumental characteristics of religion and practices aimed at self-exploration and self-knowledge that favor the subjective well-being of individuals, which could prevent drug use from becoming an alternative for dealing with conflicts in the children and young population of this region of Latin America.

**Keywords:** religion involvement; substance use problems; schoolchildren; Chile

## 1. Introduction

Substance abuse is a global public health problem. Currently, there is an increase in substance use, and, according to studies, it has been gradual in most countries (Badr et al. 2014). In 2015, around 29.5 million people, 0.6% of the adult global population, had problematic use and suffered from substance abuse disorders, including dependence (United Nations Office on Drugs and Crime 2017). In the same year, it was reported that, globally, 240 million people suffered from alcohol use disorder, 1 billion smoked tobacco products, and 15 million used injected drugs (Gowing et al. 2015).

The effects of problematic substance use involve a significant and progressive deterioration of the physical, psychological, and social development of individuals, which depends on the frequency and quantity of use, as well as on the type of drug used. However, health problems, such as digestive disorders; alterations in the central nervous system; respiratory,



neurological, cardiovascular, or cerebrovascular diseases; and sexually transmitted diseases (viral hepatitis, tuberculosis, HIV/AIDS, human papilloma virus, etc.) tend to be observed in most users (WHO 2021).

Moreover, mental health problems tend to be observed, such as difficulty in memorizing and learning, alterations in mood and perception, decreased attention and coordination, anxiety, depression, sleep disturbances, and eating behavior problems, among others. At the same time, negative effects on interpersonal development have been observed in individuals with problematic substance use, mainly exacerbated aggressive impulses that translate into all kinds of transgressions to the people around them, and risky sexual and criminal behavior (Calvete and Estévez 2009).

However, by the mid-twentieth century, research related to spirituality, religion, and health started to grow (Quiceno and Vinaccia 2009). It is important to highlight that religious involvement is a different phenomenon from what we know as spirituality, since the latter is linked to reflective processes that evoke the understanding, acceptance, and integration of human experience in temporality, as well as the attribution of meanings and the constant search for meaning from which converges an opening of consciousness and an existential vision of oneself, others, and the world more broadly (Chaves and Aranha 2015).

Religious involvement, however, involves convictions and practices sustained in faith, where the existence and scope of the divine (God) in people's lives justify the following of values, principles, attitudinal positionings, and norms of individual and social behavior (Melo et al. 2015). It should be noted that both spirituality and religious involvement have proven to be sources of well-being for individuals at different moments in life (Carvalho et al. 2015; Melo et al. 2015).

The findings on the role of religious involvement in people's psychological well-being prompted the development of a model of religious coping, where different strategies can be distinguished that allow individuals to put into practice and confirm religious convictions or beliefs from which they feel capable of solving complex challenges or problems in their lives; among these strategies are the following: (1) Self-directed style (individuals recognize and value their own abilities or merits in problem solving more than God's power or favor), (2) Avoidant style (individuals recognize and value God's power or favor in problem solving more than their own abilities, and (3) Collaborative style (both God's power or favor and individuals' abilities or merits are precursors to problem solving) (Pargament 1997).

In relation to the above, evidence has been obtained on the orientations that individuals may choose, distinguishing a personal intrinsic orientation, where the described religious practices have an inspirational or motivational sense and are not a means to achieve personal or social goals, and a social extrinsic orientation, where the described religious practices have an instrumental sense to achieve personal or social goals (Rivera-Ledesma et al. 2016).

Each of the aforementioned religious coping strategies reveals in individuals the level of self-efficacy, locus of control, autonomy, and self-confidence in the face of crises, problems, or life challenges, an aspect that can positively or negatively affect their degree of well-being or mental health (Muñoz and Moreno 2003). Empirical evidence shows that spiritual activities and/or religious involvement are associated with a decrease in depression, anxiety, and substance use (Abdollahi and Abu Talib 2015). Therefore, these could be protective factors against stressful conditions and contribute to mental health.

Results from previous research show that a purposeful and meaningful life, spiritual activities such as praying, and participating in religious communities prevent high substance use (Morjaria and Orford 2002). Similarly, studies support the fundamental influence of spiritual orientation and experiences on the etiology of substance use, its course period, and consequences (White 2008).

Studies with Latin American adolescents have revealed that religious involvement decreases the likelihood of problematic substance use, noting that participation in youth groups led by spiritual guides who promote religious values and guidelines oriented

toward the reconciliation of young people with themselves and others favors positive coping with problems where the need to experiment, maintain, or relapse into drug use tends to be mitigated (Ai and Lee 2018).

Other findings in San Salvador youth and adults suggest that religious involvement may not only act as a protective factor against drug use but also against the risk of belonging to gangs, i.e., criminal or delinquent groups (Salas-Wright et al. 2013). Other scientific findings in the Latin American population consider that religious involvement tends to act as a protective factor against the search for sensations through drugs; i.e., it can reduce the likelihood of adolescents and young adults trying substances and initiating problematic substance use (Chen et al. 2004).

The number of adolescents in Latin America and the Caribbean between the ages of 10 and 19 is 111,734,000, a figure that represents approximately 18% of the total population of the region, being considered an important risk group due to the critical conditions of their social, economic, and political contexts associated with inequality and poverty, family dysfunctionality, segregation or discrimination, violence, drug trafficking, and limited access to basic services (such as housing, employment, health, education, and social security), a scenario that favors the emergence of mental health problems (among these addictions) and involvement in criminal or delinquent groups (World Health Organization 2017).

The reality of Latin American adolescents reflects important crises and gaps that the world society faces, which is why it is urgent and necessary to develop research on this population that will allow the states to know what strategies are capable of safeguarding the psychological well-being of young people despite the adversity of their contexts, and that is precisely the importance of studies such as this one in Latin America.

In Chile, the small amount of evidence available to date suggests that, in adolescents and young adults, religious coping decreases the probability of substance use, especially if their upbringing is accompanied by a good parental performance of parents or caregivers, as well as living in a place where cultural identity, good communication, and positive affection are promoted; however, there is no clarity about what types of religious strategies (individual or collective) are the most conducive to distancing young people from drug use (Belintxon et al. 2020).

According to the characterization of the substance consumption of children and youth in Chile, the Annual Technical Evaluation Report on the treatment program for children and adolescents with a problematic consumption of alcohol and other drugs, issued by Servicio Nacional para la Prevención y Rehabilitación del Consumo de Drogas y Alcohol (SENDA) in 2018, states that marijuana (60.1%) is the drug that most frequently motivates admission to rehabilitation treatments, followed by cocaine (20.3%) and alcohol (10.8%). The use of sedatives, such as diazepam or clonazepam (7.5%), inhalable drugs such as neoprene GHB nitrous oxide (0.9%), opiates (0.3%), and amphetamine stimulants (0.1%) is also observed. According to the records of the same report, the age range with the highest consumption figures is between 16 and 20 years old (79%); however, the percentage of adolescents between 10 and 15 years old who report drug use (20%) is considered equally worrying. With respect to marijuana, it is warned that its consumption usually appears in early ages and increases with age; by contrast to the use of sedatives, an early consumption that diminishes with age has been observed for the use of other substances, such as cocaine and alcohol (SENDA 2017).

Specifically, in the Arica and Parinacota regions (north of Chile), SENDA reported that marijuana consumption in a previous year was 12.5% in the 12- to 34-year-old group and 2.0% in the 35- to 64-year-old group in 2016; cocaine consumption in a previous year was 1.5% for the first group and no case was reported for the second group. Additionally, the incidence has increased systematically since 2012 according to the same study (SENDA 2017).

Given that addictions are described as chronic and progressive disorders with high costs to the patient, their family, and society, it is necessary to analyze a number of factors,

such as those that could be present at the time that consumption behaviors are initiated. This area is poorly addressed in Chile, where drug consumption starts from very early ages with dramatic consequences.

The aim of this study is to analyze the relationship between religious involvement and substance use behaviors in students from the northern regions of Arica and Parinacota, a fraction of society in which a greater consumption of substances has been demonstrated in children and young people.

The main hypothesis that this study tests is that the greater the religious involvement experienced by children and adolescents in this region of Latin America, the lower the risk behavior for substance use.

In addition, we expect to find evidence on the role of ethnic identity in substance use, since this variable tends to behave as a determinant of mental health in Latin American adolescents and young adults with the existence of 400 indigenous groups in the region, representing approximately 45–48 million people, where at least 17% are in conditions of extreme poverty. Specifically, the Latin American indigenous adolescent population has low school enrollment rates, high school failure and dropout rates, a higher prevalence of mental health problems, high levels of discrimination and social exclusion, and limited access to basic services (such as health, housing, employment, social security, and education) (World Health Organization 2017), which is why it is worth including ethnic identity as a variable in all studies on mental health in the Latin American child and young population.

## 2. Materials and Methods

This is a non-experimental study, since variables are not manipulated. The design is ex post facto retrospective with a single group due to the fact that all variables are measured in a single moment and the aim is to establish causal relationships between variables based on theory (Hernández et al. 2010).

This study was carried out in 2018, prior to the pandemic.

### 2.1. Participants

The sample included 2313 schoolchildren between the fourth year of primary school and the last year of secondary school, who were from different educational establishments in the city of Arica, in the far north of Chile. The age range was 12–18 years. This is the age range considered by the questionnaire used in this study. There was no agreement about a cutoff of missing data, but 25% of missing data per subject can be considered moderate (Dong and Peng 2013; Schlomer et al. 2010). It was decided to remove cases with more than this percentage, and the range of missing data per item was finally less than 5%. Mean imputation was used for these items, considering that there was no difference present with other imputation methods under this percentage (Cheema 2014). The selection was based on availability; however, an attempt was made to incorporate schools from all socioeconomic levels.

### 2.2. Instruments

Demographic data: These were collected using an ad hoc scale that included questions on sex, classes, religion, age, and ethnicity.

Child and Adolescent Evaluation System (Sistema de evaluación de niños y adolescentes SENA) (Fernández-Pinto et al. 2015): Developed by specialists in psychopathology and psychological evaluation, its purpose is to help in the detection of a wide range of emotional and behavioral problems in those aged 3–18 years old. It is worth noting that it was built and validated entirely in Spanish. Among the results that can be obtained are the following:

- Internalized problems: depression, anxiety, social anxiety, somatic complaints, post-traumatic symptomatology, and obsession–compulsion.
- Externalized problems: attention problems, fast pulse, anger control problems, aggression, defiant behavior, and antisocial behavior.

- Contextual problems: problems with family, problems with school, and problems with peers.
- Specific problems: substance use, eating behavior problems, developmental delay, learning problems, schizotypy, and unusual behavior.

Recently, Sánchez-Sánchez et al. (2016) found that the reliability of their subscales is above 0.7 in Spain. To date, this instrument has not been validated in Chile in children or adolescents.

In the present investigation, the substance consumption dimension was used, which consists of 6 reagents. This varies from 1 to 5 in the response options of each item, from "never or almost never" to "always or almost always". The total of each dimension is the average of the responses that constitute it, and it can vary between 1 and 5 continuously.

The substance consumption dimension addresses the consumption of alcohol or other drugs in the company of friends and in moments of leisure, for example: "When I go out to have fun, I take drugs" and "When I go out with my friends I drink alcohol". Its reliability in this study was $\alpha = 0.84$. This scale is part of the 12- to 18-year-old SENA self-report version used in this study.

Age Universal I-E-12 (Gorsuch and Venable 1983; Gorsuch and McPherson 1989): This consists of 12 items, 6 of which assess religious orientation (I); 3 assess personal intrinsic orientation (EP), where the religious practices described have an inspirational or motivational sense and are not a means to achieve personal or social goals; and 3 assess social extrinsic orientation (ES), where the religious practices described have an instrumental sense to achieve personal or social goals. The response format suggested in Maltby's (2002) adaptation was used. The response options comprise 1, (yes), 2 (sometimes), and 3 (no), and they indicate the presence or absence with which the subject associates his or her behavior. The total score ranges from 12 to 36. A higher score indicates greater religious involvement. This instrument was validated in Chile in 2020 by Flores, Caqueo-Urízar, Hernández, and Vargas, obtaining a reliability of the total scale of 0.91. Its combined intrinsic dimension is 0.91, and its social extrinsic dimension is 0.81. The reliability in this study was $\alpha = 0.92$.

Revised Multigroup Ethnic Identity Scale (MEIM-R) (Phinney and Ong 2007): This scale was translated into Spanish by Lara and Martínez-Molina (2016). Its purpose is to have an adequate measure of ethnic identity. The original version devised by Phinney (1992) has 14 items. The revised version consists of 6 items that are scored on a Likert scale with 5 options, where 1 represents "totally agree" and 5 represents "totally disagree". It has two dimensions: Exploration and Commitment. Rivas-Drake et al. (2014) highlight instruments based on all versions of MEIM as those that yield more significant results by relating ethnic identity with other outcome variables, such as well-being or symptomatology. The higher the score, the higher the ethnic identity. To date, this instrument has not been validated in Chile; however, for its use in this work, the psychometric properties obtained from international research that used it in Spanish-speaking groups, English-speaking groups, and groups with different ethnic origins were reviewed. The reliability in this study was $\alpha = 0.92$.

### 2.3. Procedure

- Approval was obtained from the Ethics Committee of the University of Tarapacá. This study is part of a larger Educational Justice Center project.
- A total of 42 educational establishments in the city of Arica were invited to participate in the study. A total of 69% agreed to participate in the study, comprising a total of 29 establishments.
- The parents were asked for their consent after explaining the purpose and scope of the research, and, subsequently, the students were asked for their consent.
- Evaluation: Each scale was administered in classrooms and lasted approximately 45 min. At least two trained interviewers were present per room to answer questions, along with the classroom teacher.

*2.4. Data Analysis*

Data are expressed as percentages or means and standard deviations. Associations between substance use (CONSU) and continuous variables (age, religious involvement, and ethnic identity) were analyzed using Spearman correlations. A comparison of consumption means across different groups (sex, religion, and ethnicity) was performed using Mann–Whitney and Kruskal–Wallis tests.

A multiple regression analysis was performed to examine variables potentially associated with substance use. First, a univariate analysis was performed to establish which variables to include in the regression equation. Then, a multivariate analysis was performed with those variables that had a significant relationship with the consumption variable based on a threshold value of $p < 0.05$. From this point on, the method of successive steps was used to define the variables that entered the regression model. The final model incorporates standardized β coefficients, which represent a change in the standard deviation of the dependent variable (substance consumption). The three dimensions of Age Universal were considered as separate entries in the models because it was possible that not all of them had the same load. The presence of collinearity between the independent variables was discarded by means of the inflated variance factor, which was <10 in all of them (Thompson et al. 2017). All statistical tests had two tails. Statistical package SPSS version 21 was used.

## 3. Results

The final sample consisted of 2313 students. The demographic characteristics are detailed in Table 1. It can be seen that the distribution of classes is quite homogeneous.

**Table 1.** Characteristics of the sample.

| Variable | % |
|:---:|:---:|
| **Gender (Female)** | 50.4 |
| Aymara | |
| Yes | 25.9 |
| No | 74.1 |
| Religion | |
| Catholic | 47.3 |
| Evangelical | 9.7 |
| Other [1] | 10.2 |
| None | 32.7 |
| Grade | |
| Primary | |
| 7th | 21.7 |
| 8th | 19.9 |
| Secondary | |
| 1st | 17.6 |
| 2nd | 15.5 |
| 3rd | 14.1 |
| 4th | 11.3 |

[1] Other religions: Jehovah's Witness, Jewish, Muslim, and other.

The descriptive analyses of the variables are presented in Table 2. The dimension of religious involvement was considered, as well as the total scale. It can be observed that substance consumption is not normally distributed, since asymmetry and kurtosis exceed the classical criteria (George and Mallery 2010), as well as those proposed by Ryu (2011). The latter suggests the use of nonparametric statistics to compare groups (Siegel and Castellan 2005).

**Table 2.** Descriptive statistics of the variables in the study.

|  | Lowest | Highest | *M* | *SD* | Asymmetry | Kurtosis |
|---|---|---|---|---|---|---|
| Age in years | 12 | 18 | 14.35 | 1.77 | 0.249 | −1.088 |
| Inv. Religion | 12 | 36 | 18.33 | 6.11 | 0.829 | −0.303 |
| ORI INT PERS | 6 | 18 | 9.68 | 3.41 | 0.666 | −0.614 |
| ORI EXT SOC | 3 | 9 | 4.04 | 1.46 | 1.403 | 1.125 |
| ORI EXT PERS | 3 | 9 | 4.62 | 1.76 | 0.871 | −0.297 |
| ID Ethnicity | 6 | 30 | 14.30 | 6.43 | 0.367 | −0.728 |
| CONSU | 1 | 5 | 1.31 | 0.59 | 2.604 | 7.540 |

Note: CONSU: substance use, Inv. Religion: religion involvement, ORI INT PERS: personal intrinsic orientation, ORI EXT SOC: extrinsic social orientation, ORI EXT PERS: extrinsic personal orientation, ID Ethnicity: ethnic identity.

The bivariate and multivariate analyses are detailed in Table 3. In the univariate analyses, the consumption variable was associated with gender, where males scored higher in substance use ($p < 0.01$); ethnicity, where children and youths not belonging to the Aymara ethnicity presented higher scores in substance use ($p < 0.01$); age ($p < 0.01$), indicating that the older the age, the higher the substance use; and the dimensions of personal intrinsic orientation ($p < 0.01$) and extrinsic social orientation ($p < 0.01$) of religious involvement. The authors decided to consider religious involvement for each dimension because each contributes to an important area of the construct.

**Table 3.** Factors associated with substance use.

|  | Univariate Analysis | | | | Multivariate Analysis | |
|---|---|---|---|---|---|---|
|  | R [b] with CONSU | Average Range | Mann–Whitney U or Kruskal–Wallis Test | *p* | β [c] Standardized | *p* |
| Gender |  |  | 620,795.5 ** | 0.001 | 0.039 * | 0.043 |
| Female | - | 1116.00 |  |  |  |  |
| Male |  | 1198.82 |  |  |  |  |
| Aymara | - |  | 475,172.5 ** | 0.003 | 0.051 ** | 0.008 |
| Yes | - | 1094.10 |  |  |  |  |
| No | - | 1178.93 |  |  |  |  |
| Religion |  |  | 8.4782 * | 0.037 | - |  |
| Catholic | - | 1130.47 |  |  |  |  |
| Evangelical | - | 1185.10 |  |  |  |  |
| Other [a] | - | 1111.88 |  |  |  |  |
| None | - | 1201.22 |  |  |  |  |
| Age in years | 0.466 ** |  |  | 0.000 | 0.375 ** | 0.000 |
| ORI INT PERS | −0.074 ** | - | - | 0.000 | −0.045 * | 0.021 |
| ORI EXT SOC | −0.036 | - | - | 0.086 | - |  |
| ORI EXT PERS | −0.070 ** | - | - | 0.001 | - |  |
| ID Ethnic | −0.052 ** | - | - | 0.013 | - |  |

ORI INT PERS: personal intrinsic orientation, ORI EXT SOC: extrinsic social orientation, ORI EXT PERS: extrinsic personal orientation, ID Ethnical: ethnic identity. * $p < 0.05$; ** $p < 0.01$. [a] Jewish, Muslim, Mormon, Orthodox, Jehovah's Witness, and Indigenous Spirituality. [b] R Spearman correlation coefficient. [c] β standardized Beta coefficient (β represents the change in the standard deviation in the CONSU score resulting from the change in a standard deviation of the independent variable).

In the multivariate analysis, all of the aforementioned variables were considered significant ($p < 0.05$) to be entered into the regression equation, but only four variables remained in the final regression model. These variables were age ($β = 0.375$, $p < 0.01$), ethnicity ($β = 0.051$, $p < 0.01$), the dimension of personal intrinsic orientation of religious involvement ($β = −0.045$, $p < 0.01$), and gender ($β = 0.039$, $p < 0.05$). This model explained 15% of the variance of consumption.

## 4. Discussion

The objective of the present study was to analyze the relational dynamics between religious participation and substance use in schoolchildren in order to find evidence about the extent of personal orientation intrinsic to religious participation, gender, age, and ethnic identity as relevant variables associated with consumption.

Before answering the aim of the study, it is necessary to highlight the fact that 32.7% of the respondents stated that they do not belong to any religion. This result is of interest and is coherent with what Chile has experienced in relation to the participation of people in some religions. In this country, 55% of those surveyed said they were Catholics—although this figure is lower than that obtained in 2008, when 69% of the population considered themselves to be of this religion—while 16% said they identified with the evangelical religion and 5% with other religions. However, the percentage of those who did not identify with any religion is higher than that of the Latin American average, reaching 24% of the surveyed population, which means a considerable increase since 2008, where this percentage only reached 11% of the total sample (Centro de Estudios Públicos 2018). Therefore, in this study, we observed a tendency not to identify with or participate in any religion, particularly among the youth population.

In particular, the intrinsic personal orientation of religious participation, which, for the purposes of this study, was understood as spirituality, could act as a protective factor for substance use when it is among the personality characteristics or attitudes of individuals whose actions seek discovery, knowledge, and understanding of themselves, promoting self-concept and harmony between beliefs and emotions (Rezende-Pinto et al. 2018; Bonelli and Koenig 2013; Edlund et al. 2010; Vaughn et al. 2016).

Studies that have obtained similar findings suggest that an intrinsic religious orientation or spirituality acts as a psychosocial moderator of the use of substances, together with the conditions of family support, and against affective and protection needs; the sense of group belonging; and personality traits associated with assertiveness, resilience, independence or self-sufficiency, optimism, openness to change, capacity of adaptation, responsibility, and proactivity (Gartland et al. 2019; Ben-David and Jonson-Reid 2017; Fisher et al. 2017). Some authors propose that an intrinsic orientation toward religion improves subjective well-being to a greater extent than an extrinsic personal orientation, given that the latter compromises the instrumental use of religious practices aimed at resolving needs or conflicts that circumstantially generate discomfort in individuals, increasing the probabilities of presenting anxious or depressive symptoms if they cannot be resolved (Ellis and Wahab 2013; Francis 2010; Pössel et al. 2011). In addition, it is estimated that the superficiality of religious beliefs and convictions usually present in both personal and social extrinsic orientations tend to be precursors of the weakening of the subjective meaning that individuals attribute to their own experiences, given that the motivational impulse is not sustained by the search for the meaning of life but by the attainment of ends, skewing the processes of deliberation or decision making in relation to their own welfare (García-Alandete et al. 2013).

In relation to ethnic identity, the results obtained in this research show similar characteristics to those in other studies, in which ethnicity, understood as the development of a collective identity that recognizes—as its own certain attributes, values, and beliefs—that other individuals or groups do not, may, or may not act as a potential protective factor for substance use, since there is evidence that this depends not only on the attribution of meanings that each group gives to the use of substances but also on the fact that individuals belong to minority groups, which, by perceiving rejection or ethnic-racial prejudice from dominant groups, tend to hide their own identity so as not to be subject to discrimination, becoming a defensive strategy that enables subjects to function and adapt to these hostile social or cultural conditions in their environment (Marsiglia et al. 2004; Lee 2005; Castle et al. 2011; Stein et al. 2014; Williams et al. 2014).

Another finding of the present study was age as a relevant variable associated with substance use, with similar results being observed in other research, where the temporal

conditions in which consumption is initiated and develops have special importance, since consumption usually appears in adolescence and tends to be an indicator of abuse and dependence toward the beginning of adulthood, increasing the probabilities of adopting antisocial and risk behaviors, along with functional alterations in cognitive development (Hernández et al. 2009; Agrawal et al. 2006; Trenz et al. 2012). In relation to the above, it is important to point out that age does not serve as an indicator of problematic consumption when it is isolated data; the evidence reveals the existence of factors related to impulsive, transgressive, and anxious personality traits; dysfunctional family conditions; experiences of mistreatment or abuse; school failure; and the lack of resources that, when accompanied by an early age of consumption, increase the probabilities in adolescents of developing addictions in adulthood (Stone et al. 2012; Piko and Kovács 2010; Caballero et al. 2010; Rueda-Jaimes et al. 2011).

Finally, in relation to sex, the men in this sample reveal a greater tendency toward the use of substances than women do, a finding that corresponds with the results of other studies where, in addition to observing differential dynamics by sex in the problematic use and consumption of drugs, there are also differences between men and women in relation to consumption patterns, motivational impulse, and the meaning attributed to the use of substances, usually observing higher figures and more rigid and harmful patterns in the male population (Chen and Jacobson 2012; Schwinn et al. 2010). In this regard, it is estimated that society and culture tend to show a greater acceptance and normalization of consumption in men, while in women, this behavior tends to represent irreverence, offense, and disgrace to good customs, observing lower figures and consumption patterns less rigid and harmful to health in the female population in relation to frequency, quantity, or dose and types of substances (Leatherdale and Burkhalter 2012; Wu et al. 2010; Brody et al. 2012). Some authors suggest that the stigmatization of women who use drugs tends to increase the likelihood of developing depression, given the tendency of women to hide the problem of addiction and isolate themselves socially (Chen et al. 2011).

Problematic substance use is a public health problem that poses a significant risk when it begins at an early age, as it increases the likelihood of developing dependence, as well as affecting the neuropsychological development of young people (Jones et al. 2019; Rioux et al. 2018; Bjorkenstam et al. 2018). Therefore, prevention programs must commit with special importance to the timely discovery of not only the signs or symptoms related to physiological or psychological functional alterations but also the psychosocial factors that favor or weaken drug use, meaning any type of substance with addictive potential, depending on the consumption pattern (frequency, dose, and route of administration), the socio-cultural context, and the individual characteristics of the subjects (Arnaud et al. 2019; Broning et al. 2012).

It is important to consider the limitations of this study, which are related to the difficulty in explaining substance use in the adolescent population since it is not plausible to interpret the findings of this research in terms of "causal relationships". It is suggested that age, ethnicity, gender, and intrinsic religious orientation be considered only as variables associated with substance use according to the results presented.

Another limitation of this study is the absence of other psychosocial variables of consumption in students, such as personality traits; perceived discrimination; and experiences of mistreatment, abuse, or trauma. It is suggested that these be incorporated into future research. A third limitation is that the SENA questionnaire used is still in the process of validation in Chile. Finally, it is necessary to mention that the statistical analyses performed do not allow us to discriminate between alcohol consumption and the consumption of other substances.

## 5. Conclusions

After analyzing the relationship between religious participation and substance use in adolescent students attending school in northern Chile, it was found that an intrinsic religious orientation or spirituality is a relevant variable associated with consumption due

to its non-instrumental characteristics of religion and practices aimed at self-exploration and self-knowledge that favor the subjective well-being of individuals, which could prevent drug use from becoming an alternative for dealing with conflicts. Other relevant findings were sex and age, with a greater tendency of substance use in men and early initiation of use as a potential predictor of the emergence of substance use and dependence disorders in adulthood. Finally, it was noted that ethnic participation may or may not act as a potential protective factor depending on the collective attribution of meaning and significance against substance use and the membership of individuals in culturally or socially segregated minority groups.

**Author Contributions:** Conceptualization, A.C.-U., A.U., E.P., J.F. and M.I.; methodology, A.C.-U., A.U., E.P., J.F., C.E.-S., K.R. and G.Z.; software, A.C.-U., A.U., E.P., J.F. and M.I.; validation, A.C.-U., A.U., E.P., C.E.-S., J.F. and M.I.; formal analysis, A.C.-U., A.U., E.P., J.F., C.E.-S., K.R. and G.Z.; investigation, A.C.-U., A.U., E.P., J.F., C.E.-S., K.R. and G.Z.; resources, A.C.-U., A.U., J.F. and M.I; data curation, A.C.-U., A.U., E.P., J.F., C.E.-S., K.R. and G.Z.; writing—original draft preparation, A.C.-U., A.U., E.P., J.F. and M.I.; writing—review and editing, A.C.-U., A.U., E.P., J.F. and M.I.; visualization, A.C.-U., A.U. and M.I.; supervision, A.C.-U., A.U., E.P., J.F. and M.I.; project administration, A.C.-U., A.U., J.F. and M.I.; funding acquisition, A.C.-U. All authors have read and agreed to the published version of the manuscript.

**Funding:** This research was funded by Agencia Nacional de Investigación y Desarrollo de Chile (ANID), grant ANID PIA CIE 160007.

**Institutional Review Board Statement:** The research was approved on September, 20th of 2017 by the Scientific Ethics Committee of the University of Tarapacá (Approval Certificate NO. 26.2017), whose resolution accredits that all data collection materials, methodological procedures and criteria for confidentiality and protection of participants comply with current regulations under Law 20.120 of the Organic Regulations of the Ministry of Health of Chile and in accordance with the protocols of the 1964 Helsinki Convention on Scientific Research Involving Human Subjects.

**Informed Consent Statement:** Informed consent was obtained from all subjects involved in the study.

**Data Availability Statement:** The data presented in this study are available on request from the corresponding author.

**Conflicts of Interest:** The authors declare no conflict of interest.

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
