# Peer review of "Religion Involvement and Substance Use Problems in Schoolchildren in Northern Chile"

_religions, doi:10.3390/rel13050442_

Round 1

Reviewer 1 Report

The article is interesting but needs some corrections:

1.In line 26 it says "Faith", it is difficult to say why with a capital letter.

2.In line 78 and following, the author writes about research conducted in Latin America. It should be noted, however, that the term Latin America is extremely capacious and includes both poor and rich countries with religious diversity. Moreover, in the paragraph beginning with line 84, he refers to an article that does not deal with Latin America as such.

The text therefore requires correction and some clarification.

3.The hypothesis contained in line 147 needs to be highlighted.

4.In the methodology section, the reviewer does not see information about the time when the research was conducted. The evaluation of the discussion and conclusions depends to some extent on this information.

5. According to the reviewer, the discussion did not take into account the fact that 32.7% of the respondents (Table 1) stated that they do not belong to any religion.  

It is not easy to assess whether a text needs minor or major changing. 

Reviewer 2 Report

The article aims to study the relationship between religious involvement and substance abuse in adolescents from the Arica and Parinacota regions of Chile.  I consider that the article deals with an important topic in a very clear way and has scientific quality to be published in the journal.

I congratulate the authors for their work.

I only have a couple of minimal suggestions:

There is a mistake in the spelling of "with". Correct whit (line 171)
I suggest the authors shorten some of the paragraphs so that the reading is more fluid and understandable (eg. lines 105-112)

Thank you very much for this contribution

Reviewer 3 Report

A very interesting paper, with a significative sample and clear in its development. Anyway, there are some comments and suggestions I would like to make for the authors in order to help for paper´s improvement:

  • Abstract needs to be reviewed by the authors, there is no relevant information about the research in it and it´s basic to show the power of the study.
  • Introduction is correct, but it would be much more better if the explanation about Substance abuse (lines 56-77) would be the beginning of the paper.
  • I highly recommend to change in Prevention papers the concept "Substance abuse" to "Substance use". i know there is a big discussion in scientific forum about what is "the proper one", but about Prevention of substances is very difficult to talk about "abuse" in first step.
  • It´s necessary to include the aims and hypotheses of the study in the last part on Intro or at the start of Method (the aim is expressed in Discussion), but I think it would improve the clarity of the paper
  • Results are very interesting, but data looks a little bit confuse. The table 2& 3 need to be better descripted, at less the significative data, in order to connect with the Discussion.
  • About Results and Discussion, it would be wonderful if there is any kind of analysis about combination of factors or the weight of factors in the prevention. It´s a pity to have a big sample and not tu use it going deeper in the variables.
  • In lines 384-389, the authors say "After analyzing the relationship between religious participation and substance use in adolescent students attending school in northern Chile, it was found that intrinsic religious orientation or spirituality is a relevant variable associated with consumption due to its non-instrumental characteristics of religion and practices aimed at self-exploration andself-knowledge that favor the subjective well-being of individuals, which could prevent drug use from becoming an alternative for dealing with conflicts in their lives". In my humble opinion, there are parts of this conclusion that are not sustained by the data of the study, so I highly recommend to conclude only what your data can sustained

Round 2

Reviewer 3 Report

Lot of thanks to the authors for their efforts to include the recommendations and suggestions into the text. Congratulations, you have significatively improved the quality of the paper